# Technical note: Long-term persistence loss of urban streams as a metric for catchment classification

Dusan Jovanovic[1], Tijana Jovanovic[2], Alfonso Mejía[2], Jon Hathaway[3], Edoardo Daly[1]

[1]Department of Civil Engineering, Monash University, Melbourne, 3800, VIC, Australia
[2]Department of Civil and Environmental Engineering, The Pennsylvania State University, University Park, 16802, PA, USA
[3]Department of Civil and Environmental Engineering, The University of Tennessee, Knoxville, 37996, TN, USA

*Correspondence to*: Edoardo Daly (edoardo.daly@monash.edu)

**Abstract.**

Urbanisation has been associated with a reduction in the long-term correlation within a streamflow series, quantified by the Hurst exponent ($H$). This presents an opportunity to use the $H$ exponent as an index for the classification of catchments on a scale from natural to urbanised conditions. However, before using the $H$ exponent as a general index, the relationship between this exponent and level of urbanisation needs to be further examined and verified on catchments with different levels of imperviousness and from different climatic regions. In this study, the $H$ exponent is estimated for 38 (deseasonalized) mean daily runoff time series, 22 from the USA and 16 from Australia, using the traditional rescaled-range statistic (R/S) and the more advanced multi-fractal detrended fluctuation analysis (MF-DFA). Relationships between $H$ and catchment imperviousness, catchment size, annual rainfall and specific mean discharge were investigated. No clear relationship with catchment area was found, and a weak negative relationship with annual rainfall and specific mean streamflow was found only when the R/S method was used. Conversely, both methods showed decreasing values of $H$ as catchment imperviousness increased. The $H$ exponent decreased from around 1.0 for catchments in natural conditions to around 0.6 for highly urbanised catchments. Three significantly different ranges of $H$ exponents were identified, allowing catchments to be parsed into groups with imperviousness lower than 5% (natural), catchments with imperviousness between 5 and 15 % (peri-urban), and catchments with imperviousness larger than 15% (urban). The $H$ exponent thus represents a useful metric to quantitatively assess the impact of catchment imperviousness on streamflow regime.

## 1 Introduction

The increase in the degree of urbanization of a catchment is known to adversely influence streamflow quantity and quality (Paul and Meyer, 2001; O'Driscoll et al., 2010; Fletcher et al., 2013). Larger peak-flows and reduced times to peak are well known characteristics of urban streams (Leopold, 1968), while lower baseflow rates are often but not always observed (Hamel et al., 2015). At the same time, the larger flows discharged in urban streams increase the export of nutrients and pollutants, and cause a decrease in biodiversity with associated detrimental effects to the health of riparian ecosystems (Walsh et al., 2005b). Although traditionally focused on peak flows, the management of urban stormwater runoff is now often directed toward restoration of the overall streamflow regime, which includes the dynamic range of streamflow fluctuations (see Mejía et al., 2014). Therefore, the management of urban streams has been increasingly focused on controlling urban stormwater runoff with the intent to limit the streamflow during floods while also maintaining adequate flow rates during dry periods (Debusk

et al., 2011; Burns et al., 2012). This management strategy is often linked to the natural flow paradigm (Poff, 1997), which advocates the maintenance of streams as close as possible to their natural regime.

Accordingly, streams affected by urbanization should be managed to achieve streamflow regimes similar to predevelopment conditions. It is thus important to quantify the effect of urbanization on streamflow and systematically classify urban catchments according to their level of development (e.g., McDonnell and Woods, 2004; Wagener et al., 2007; Sivakumar and Singh, 2012). A number of indexes are available for assessing the changes in hydrological regimes of rivers and streams (Olden and Poff, 2003; Poff et al., 2010); however, only a subgroup has been tested for application to a specific flow component (baseflow) in urbanized catchments (Hamel et al., 2015). These indexes are often grouped depending on the information they give about certain flow characteristics, such as magnitude, frequency, and duration. The selection of appropriate metrics that are applicable to different locations and climatic conditions is still an unresolved issue, with statistical methods often employed to avoid the use of metrics which provide redundant information on the streamflow regime (e.g., Olden and Poff, 2003; Poff et al., 2010; Hamel et al., 2015). Despite these efforts, it is still difficult to define a single streamflow metric that can summarize where a catchment lies across the spectrum of conditions from natural to fully urbanized.

Because urbanization facilitates the rapid delivery of precipitation to streams due to impervious areas and the direct connection between land and stream via drainage systems (Hamel et al., 2013; Epps and Hathaway, *in press*), the resulting flashier hydrographs and the lower baseflow rates in urban areas have been associated with a possible reduction in long-term correlation (Jovanovic et al., 2016; Kim et al., 2016). A measure of long-term correlation is represented by the Hurst exponent (e.g., Beran, 1994), which has been largely used to characterized streamflow time series (Hurst, 1951; Mandelbrot and Wallis, 1969; Montanari et al., 1997; Koutsoyiannis, 2002; Szolgayova et al., 2014; Jovanovic et al., 2016; Ausloos et al., 2017). The Hurst exponent tends to 0.5 when an aggregated signal converges to white noise, while values larger than 0.5, as commonly found for streamflow series, are associated with persistent processes (e.g., Serinaldi, 2010). Because urbanization, with the quicker delivery of stormwater runoff to streams, is likely to break the long-term correlation often found in rural and natural streams, one would expect the Hurst exponent of urban streams to be closer to 0.5 when compared to rural and natural streams. Following this hypothesis, Jovanovic et al. (2016) and Kim et al. (2016) reported a decrease in values of the Hurst exponent with an increase in catchment imperviousness. Thus, although not listed in the large number of metrics often used to classify streams, the Hurst exponent has the potential to quantify the impact of urbanization on stream hydrology. However, there is a need to further examine the relationship between the Hurst exponent and the level of urbanization of a catchment by 1) including more catchments into the analysis in order to better estimate the relationship, and 2) include catchments from different geographic and climatic regions to explore whether the observed relationship is site-independent. The assumption that catchments with lower degrees of urbanization present long-term persistence needs also to be validated across a spectrum of catchments. The aim of this study is to investigate the utility of the Hurst exponent to estimate the impacts of urbanization on stream hydrology, thereby providing a general index for the classification of different levels of catchment urbanization.

**2 Methods**

**2.1 Estimation of long-term dependence within a daily streamflow time series**

Two methods were applied to estimate the Hurst exponent, $H$. The first method is the oldest and well-known rescaled range statistics (R/S) (Hurst, 1951) as proposed by Mandelbrot and Wallis (1969). Although this method has limitations compared to other more sophisticated estimations of scaling indexes (Montanari et al., 1997; Szolgayova et al., 2014), it is rather simple and thus popular to detect long-term persistence by way of estimating $H$. The second method is multifractal detrended fluctuation analysis (MF-DFA) (Kantelhardt et al., 2002).

For both methods, the analysis starts by creating from the original series of length $L$ the fluctuation series, $\phi_j$ ($j = 1, .., L$), which does not have seasonal cycles. Seasonal cycles are approximately removed from the original series by subtracting the calendar day mean and dividing by the calendar day standard deviation (Kantelhardt et al., 2002; Szolgayova et al., 2014). The R/S and the MF-DFA are then applied to the series $\phi_j$.

The details of the R/S method can be found in Weron (2002) and Szolgayova et al. (2014). The time series $\phi_j$ is divided into $D$ subseries of length $N$. For each of the $m$ subseries ($m = 1, ..., D$), the following calculations are performed:

1) The mean $E_m$ and standard deviation $S_m$ of the subseries are calculated.

2) The subseries, $\phi_{i,m}$, is normalized by subtracting the sample mean to obtain (Equation 1):

$$X_{i,m} = \phi_{i,m} - E_m$$

(1)

for $i = 1, ..., N$.

3) Cumulative time series are created as (Equation 2):

$$Y_{i,m} = \sum_{j=1}^{i} X_{j,m}$$

(2)

for $i = 1, ..., N$.

4) The range is calculated as (Equation 3):

$$R_m = \max\{Y_{1,m}, ..., Y_{n,m}\} - \min\{Y_{1,m}, ..., Y_{n,m}\}$$

(3)

and then rescaled as $R_m/S_m$.

Once these steps have been repeated for the $m$ subseries with length $N$, the mean value of the rescaled range for all subseries of length $N$ is calculated as (Equation 4):

$$(R/S)_N = \frac{1}{D}\sum_{m=1}^{D} R_m/S_m.$$

(4)

These operations are performed for different values of $N$. The minimum length of the subseries used in this study was 10 days, while the maximum length was ¼ of the total length of the available time series. Since the $R/S$ statistics asymptotically follows the relationship $(R/S)_N \sim cN^H$ (Mandelbrot and Wallis, 1969), the value of $H$ is estimated as the slope of the linear least squares regression (Equation 5):

$$\log(R/S)_N = \log c + H \log N.$$
(5)

The details of the MF-DFA method can be found in, e.g., Jovanovic et al. (2016). The following calculations are performed on the time series $\phi_j$:

1) The so-called profile value at $i$ of the fluctuation series is created as (Equation 6):

$$W(i) = \sum_{k=1}^{i} \phi_k.$$
(6)

The profile is segmented into non-overlapping intervals of duration $s$ (Equation 7):

$$N_s = int(N/s)$$
(7)

2) Local trends from each $N_s$ segment are removed by fitting a polynomial trend to each segment and removing it from the series. This permits the calculation of the fluctuation function for each segment as (Equation 8):

$$F^2(v,s) = \frac{1}{s} \sum_{i=1}^{s} [W(vs+i) - p_{n,v}(i)]^2,$$
(8)

where $v$ indexes the different segments $v = 0, \ldots, N_s - 1$ and $p_{n,v}(i)$ is the value at $i$ of the polynomial fit of order $n$ in segment $v$.

3) Since the record length $N$ is unlikely a multiple of $s$, to avoid disregarding that portion of the record, the fluctuation function is also computed starting from the end of the record as (Equation 9):

$$F^2(v,s) = \frac{1}{s} \sum_{i=1}^{s} [W(N - (v - N_s + 1)s + i) - p_{n,v}(i)]^2.$$
(9)

To find the overall $q^{th}$ order fluctuation for the timescale $s$, the fluctuations from each segment are averaged such that

$$F_q(s) = \left\{ \frac{1}{2N_s} \sum_{i=0}^{2N_s-1} [F^2(v,s)]^{q/2} \right\}^{1/q}.$$

10)

When the time series has a long-term power-law correlation, one expects that $F_q(s) \sim s^{h(q)}$, where $h(q)$ is the generalized Hurst exponent (Kantelhardt et al., 2002). In the case q =1, $h(1)$ corresponds to the Hurst exponent from the R/S analysis (Kantelhardt et al., 2006). To obtain the scaling exponent $h(q)$, $F_q(s)$ is plotted as a function of $s$ in log-log space and a line is fitted to the data using the least square regression. In this study, *q* was 1 and the order of the polynomial fitting function was 5.

The values of *H* obtained for different streamflow series were then grouped according to the level of impervious cover (i.e. level of urbanization of catchments) to select natural, peri-urban, and urban catchments. A two sample t-test and Wilcoxon rank-sum test was applied to test the difference between the empirical distributions of the Hurst exponents for these three different groups of catchments.

Given the uncertainty in the estimation of *H*, especially when streamflow series are shorter than twenty years, the Pearson autocorrelation function of the series was calculated and the autocorrelation coefficient at 1-day delay (i.e., lag-1 Pearson autocorrelation) was selected as a metric characterizing the persistence of the series. The values of the lag-1 autocorrelation coefficient where then related to levels of catchment imperviousness as done for the *H* exponent.

**2.2 Data**

Thirty eight time series of mean daily streamflows were used for this study, as detailed in Table 1. Twenty two streamflow gauging stations were located in the USA and sixteen were in Australia. The US stations were located in the metropolitan areas of the cities of Baltimore, Philadelphia and Washington, DC, and the Australian stations were located in the metropolitan area of the city of Melbourne. The locations of these monitoring stations within the USA and Australia were visualized on maps in Jovanovic et al. (2016) and Hamel et al. (2015), respectively. The streamflow series were not affected by large structures, such as dams and reservoirs, and flow in most catchments was driven by climatic conditions and catchment characteristics. Large areas of some of the catchments in the USA are agricultural land; with the exception of Lang Lang River, which is partly used for agriculture, the catchments with no degrees of impervious areas in Australia are completely forested. The mean annual rainfall for US catchments ranged from around 1000 mm to 1250 mm. The mean annual rainfall for the Australian catchments located close to the city is about 700 mm (Melbourne water station 229636A for the period 1978-2016), while some of the outlaying catchments having low to no imperviousness receive up to about 1100 mm (Melbourne water station 229111A for the period 1990-2016). The air temperature range for US catchments is from -20 to 41 °C with an average of 13 °C, while the air temperature for Australian catchments ranges from -3 to 47 °C with an average of 15 °C. According to the Koppen-Geiger classification, the climate of the US and Australian catchments is classified as Cfa and Cfb respectively.

The level of catchment urbanization for each of the stations was indicated by the fraction of total impervious cover (i.e. imperviousness, U [%]), which ranged from zero (i.e. natural conditions) to nearly 50 % (i.e. largely urbanized catchments). For the US catchments, information about impervious data can be found in Mejia et al. (2015).

Imperviousness was calculated using a combination of tax map information and areal imagery. In terms of land use, pervious land is mainly agricultural and urban green spaces, including lawns, parks and other grassed areas. Riparian corridors, if present, are likely to be forested. The majority of imperviousness consist of residential, commercial and transportation land use areas. For the Australian catchments, the percentage of impervious area were taken from Hamel et al. (2015) and were calculated using the methods in Kunapo et al. (2005). Buildings and paved areas (i.e., roads and carparks) were mapped using a geographical information system software, and the percentage of impervious areas, U, was calculated as the ratio between total impervious area and catchment area. The land use of the catchments with larger U is residential, while those with low U are mostly covered by natural forests.

## 3 Results and Discussion

The values of $H$ obtained using R/S analysis were between about 0.6 to around 0.9, while the MF-DFA analysis resulted in slightly higher values, with H larger than 0.65 (Figure 1); MF-DFA can give estimates slightly higher than 1 (Serinaldi, 2010) due to uncertainties in the estimation of the parameter H.

Regardless of the method used, the obtained values of $H$ showed a decreasing trend with the increase in imperviousness for both US and Australian catchments (Spearman correlation coefficient $\rho = -0.68$, $p<0.001$ for R/S method and $\rho = -0.77$, $p<0.001$ for MF-DFA method; Figure 1). The decreasing trend appears to stop around 15% of imperviousness; increasing the impervious fraction beyond 15% does not seem to be associated with a further decrease of the estimated values of $H$. From Figure 1, it appears that the rapid release of stormwater runoff to urban streams, which generates the well-known flashiness in urban streamflow series (e.g. Leopold, 1968; Paul and Meyer, 2001) breaks the long-term correlation within the time series, as reflected by the lower values of $H$. Conversely, the catchments in natural conditions have the highest values of $H$ due to high precipitation filtering capacity of natural landscapes.

The lag-1 autocorrelation coefficient follows a pattern similar to $H$ for the Australian catchments, but the relationship between the lag-1 autocorrelation coefficient and imperviousness is not as evident for the catchments in the USA (Figure 1). The US catchments can be quite flashy, and this may be a reason why the lag-1 autocorrelation is low irrespectively of the level of imperviousness.

This decrease in the long-term dependence of the streamflow time series has been observed in previous studies (Jovanovic et al., 2016; Kim et al., 2016). Jovanovic et al. (2016), using data from the same US catchments as the present study, showed that the temporal structure of the streamflow for the most urbanized US catchments was similar to the temporal structure of the precipitation records. This suggests that the increase in impervious cover might cause more precipitation to bypass the groundwater storage due to a reduction in infiltration potential. Furthermore, the urban stormwater network tends to enhance the delivery of water from directly connected areas to urban streams. Although direct connectivity was not explicitly considered in this study, one would expect impervious fraction and direct connectivity to be highly correlated. The elevated degree of direct connectivity in highly impervious catchments may have contributed to the decrease in the H exponent. Because new developments in both USA and Australia are increasingly implementing stormwater green infrastructure that reduces connectivity between impervious areas and urban streams, it is possible that the reduction of $H$ will become less visible in the future.

The obtained values of $H$ can be parsed into three groups according to their impervious cover fraction (Figure 1). Catchments with less than 5% impervious areas have larger values of $H$ ($0.75<H$ using R/S and $0.9<H$ using MF-DFA), while those with values of imperviousness larger than 15% are mostly related to lower values of $H$ ($0.6<H<0.75$ using R/S and $0.6<H<0.9$ using MF-DFA). This is consistent with experimental observations, which showed that imperviousness as low as 15-20% is sufficient to impact stream health and ecology (e.g., Arnold and Gibbons, 1996; Walsh et al., 2005a). Between 5% and 15% the values of $H$ appear more scattered. Therefore, Figure 1 suggests that the analysed catchments might be classified as natural (U≤5%), peri-urban (5%<U≤15%), and urban (U>15%) catchments based on the corresponding range of values of $H$. Similar classification ranges were reported in the literature previously (e.g., Hamel et al., 2015), although they were not based on a single metric quantifiable from streamflow series. The relationship between $H$ and catchment imperviousness appears to be more evident for the US catchments, as the Australian streamflow series for two highly urbanized catchments (i.e., Blind Creek @ Knox and Gardiners Creek @ Kinkora) resulted in values of $H$ comparable to peri-urban catchments, potentially due to data gaps. Nevertheless, when considered together as three different groups (i.e., natural, peri-urban and urban), the empirical distributions of $H$ corresponding to these groups are significantly different from each other ($p<0.05$, t-test and $p<0.05$, Wilcoxon rank-sum test) for both the R/S and MF-DFA methods (Figure 2). A similar classification would result from the lag-1 autocorrelation coefficient for the Australian catchments, while the lag-1 autocorrelation would not show any visible difference in the US catchments.

Because other variables and catchment attributes can affect the value of $H$, the estimated $H$ exponents and lag-1 autocorrelation coefficients were also related to the catchment area and catchment wetness.

The differences in lag-1 autocorrelation coefficients between US and Australian catchments shown in Figure 1 make it difficult to identify a general pattern in relation to other variables (Figure 3).

Previous studies identified a positive relationship between $H$ and catchment area (e.g., Mudelsee, 2007; Hirpa et al., 2010; Szolgayova et al., 2014). The increase in long-term dependence has been hypothesized as being due to the increase in storage capacity (e.g., groundwater, inundations) with increasing catchment size. However, no relationship was found between these two variables for the dataset in Table 1, as shown in Figure 3 ($\rho = 0.24$, $p = 0.15$ for R/S method and $\rho = 0.28$, $p = 0.09$ for MF-DFA method). These conflicting results may be due to the smaller size of the catchments used here compared to those in literature. For example, the size of the largest catchment used in this analysis (see Table 1) is comparable to the smallest of the catchments used by Szolgayova et al. (2014). The groundwater storage appears to be able to affect the persistence of the series of the less urbanized catchments irrespectively of the area. However, small urbanized catchments may not have sufficient water storage to influence the long-term dependence in flow time series, and an increase in imperviousness further limits the water storage capacity of the urban catchments. This may contribute to the lack of apparent relationship between the $H$ exponent and the catchment size.

Catchments wetness, as indicated by annual rainfall and specific mean streamflow, is also known to influence the Hurst exponent (Szolgayova et al., 2014). Generally, catchments with lower rainfall totals and lower specific mean streamflow are found to have higher long-term dependence due to the longer dry weather periods and consequently longer low flow periods. In this study, weak but significant negative correlations were found between the $H$ exponent and both annual rainfall ($\rho = -0.36$, $p < 0.05$) and specific mean streamflow ($\rho = -0.35$, $p < 0.05$), but only for the rescaled-range method, indicating that to some extent this holds true for smaller catchment sizes. The

links between rainfall and streamflow persistence have been explored by Jovanovic et al. (2016) for the USA catchments; they showed that the scaling properties of quickflow in the USA stations were similar to those of some rainfall stations in the same area.

However, as demonstrated above, catchment imperviousness seems to be the most influential parameter in affecting long range correlations for these sites.

## 4 Conclusions

Catchment imperviousness has long been identified as a key environmental indicator for stream health (e.g., Arnold and Gibbons, 1996; Hatt et al., 2004). In this study, the percent imperviousness of urban catchments was also related to the loss of long-term persistence in the streamflow series as quantified by the Hurst exponent.

Streamflow data series from catchments in the USA (Jovanovic et al., 2016) were combined with data from catchments in Australia (Hamel et al., 2015) to cover a large range of imperviousness, from catchments used for water supply in natural conditions to those that were heavily urbanized.

The Hurst exponent, calculated using both the rescaled-range statistics (a simpler method) and the more sophisticated de-trended fluctuation analysis, decreased from values around 1 in natural catchments to values

around 0.6 for highly urbanized catchments. The values of the Hurst exponent, which is now not listed in the large range of metrics often calculated to characterise river regimes (e.g., Poff et al., 2010; Hamel et al., 2015), were conducive to the classification of the catchments as natural, peri-urban, and urban. That is, the Hurst exponent showed fairly visible breakpoints between these catchment types and has been shown here to be an effective and easy-to-estimate metric to distinguish different levels of catchment urbanization. The range of imperviousness

associated with this classification scheme is comparable to ranges developed via other metrics in literature, but was obtained herein using a metric easily calculated from daily streamflow data that is routinely collected by water agencies. There are obvious benefits to such a method, which can be applied without the need to use expensive and time consuming water quality and biological measurements. A downside of the method is that the estimation of the H exponent requires long time series (at least between 15 to 20 years; e.g., Koutsoyiannis (2013),

Dimitriadis and Koutsoyiannis (2015)); the method is thus not usable to determine the benefit of restoration activities on the short term. Additionally, the analyses presented here did not use any adjustment to correct for bias in the estimation of the Hurst exponent; this adjustment would in general lead to higher values of estimated H (H<1).

With the increasing availability of streamflow and rainfall data from water management agencies, it should be

possible to relate the scaling properties of rainfall and streamflow as well as detecting the effect of urbanization on rainfall patterns around cities. This was not possible using the data of this study because of the lack of rainfall stations corresponding to the flow stations and the availability of series too short to detect changes in rainfall patterns.

### Acknowledgements

The data for Australian catchments used in this study can be found at the Victorian water agency website - Melbourne Water (www.melbournewater.com.au). The streamflow data for the USA catchments can be found at the United States Geological Survey (USGS) website (www.usgs.gov), while the precipitation data can be found

at the National Oceanic and Atmospheric Administration (NOAA) website (www.noaa.gov). Edoardo Daly thanks the Faculty of Engineering at Monash University for supporting his Outside Study Program and the Department of Civil, Environmental and Architectural Engineering at the University of Padova, Italy, for hosting him when the study was finalized. The authors are grateful to A. Montanari, P. S. C. Rao, L. E. Bertassello, and P. Dimitriadis for thoroughly reviewing the paper.

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

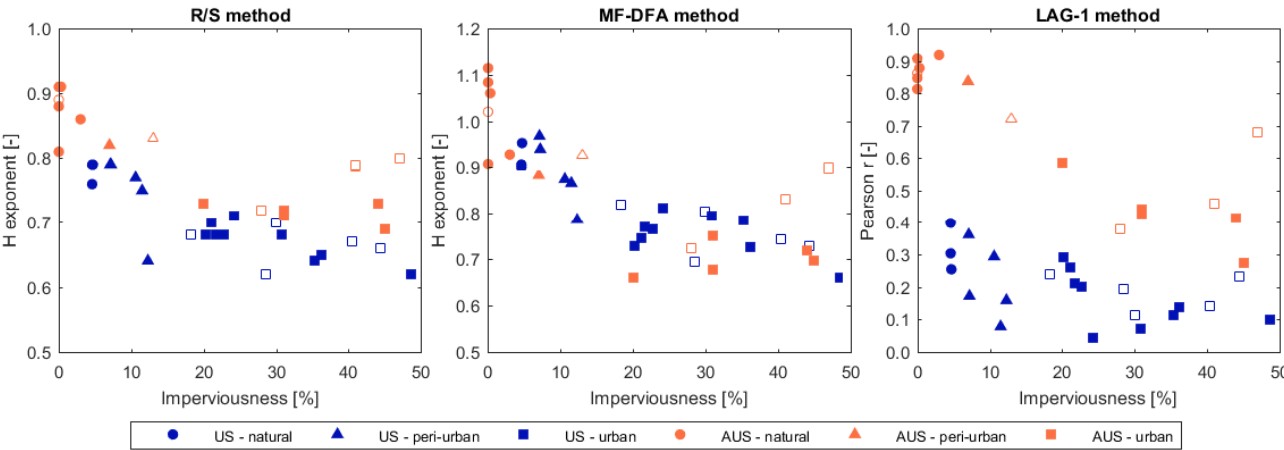

**Figure 1. Relationship between the percentage of catchment impervious area and the H-exponent of the streamflow series estimated with R/S method (left) and MF-DFA method (middle), and lag-1 Pearson's autocorrelation coefficient (r) (right) for catchments in the USA and Australia. Time series with gaps longer than 200 consecutive days have empty markers.**

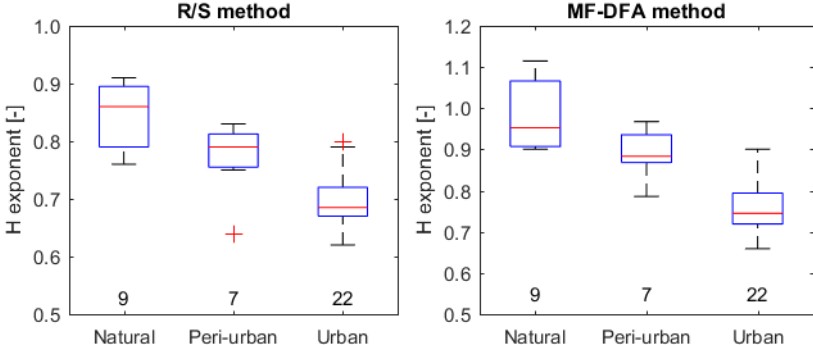

**Figure 2. Empirical probability distributions of the H-exponent of the streamflow series estimated with R/S method (left) and with MF-DFA method (right) grouped into natural (U<5%), peri-urban (U=5-15%) and urban (U >15%) catchments in the USA and Australia. Numbers below the boxplots indicate number of data points used to generate the empirical distributions.**

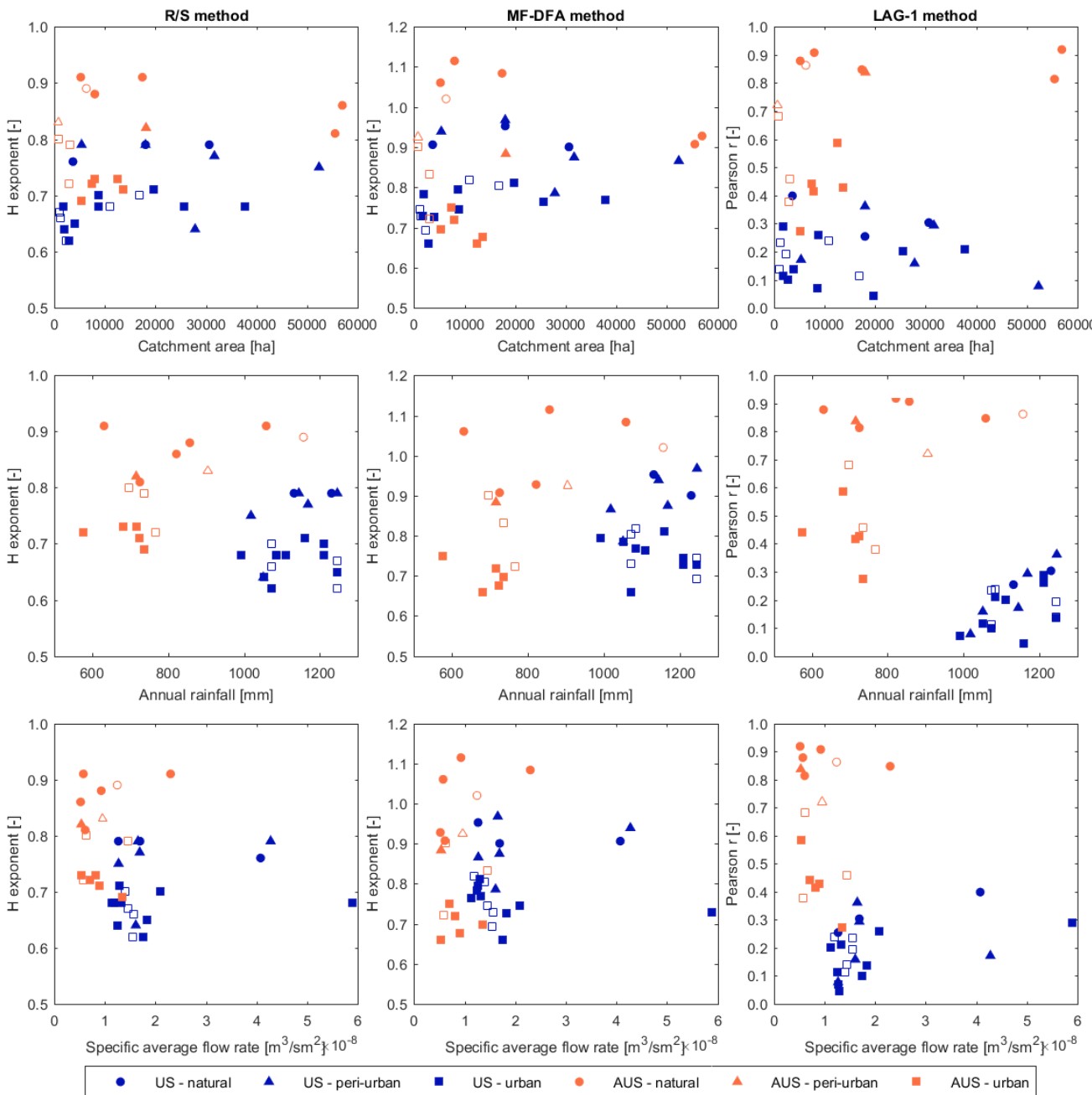

**Figure 3. Relationship between the H-exponent of the streamflow series (estimated with R/S method (left) and with MF-DFA method (middle)), and lag-1 Pearson autocorrelation coefficient (right) with the catchment size (top), annual rainfall (middle) and average flow rate per unit area (bottom) for catchments in the USA and Australia. Time series with gaps longer than 200 consecutive days have empty markers.**

**Table 1.** US and Australian streamflow monitoring stations with monitoring period, number of missing days, fraction of impervious area, and total catchment area. Grey cells indicate monitoring stations with a gaps in time series longer than 200 consecutive days.

| No. | Name | Gauge ID | Location | From – To (number of years) | Total missing days (consecutive missing days) | Impervious fraction U [%] | Catchment Area [ha] |
|---|---|---|---|---|---|---|---|
| 1 | French Creek | 1472157 | US | 1968-2013 (45) | / | 4.6 | 15310 |
| 2 | Hunting Creek | 1641000 | US | 1949-1992 (43) | / | 4.6 | 1840 |
| 3 | Patuxent River | 1591000 | US | 1944-2013 (69) | / | 4.7 | 9010 |
| 4 | Winters Run | 1581700 | US | 1967-2013 (46) | / | 7.1 | 9010 |
| 5 | Raccoon Creek | 1477120 | US | 1966-2014 (48) | / | 7.2 | 2690 |
| 6 | Chester Creek | 1477000 | US | 1931-2013 (81) | / | 10.6 | 15820 |
| 7 | Seneca Creek | 1645000 | US | 1930-2013 (83) | / | 11.5 | 26160 |
| 8 | Skippack Creek | 1473120 | US | 1966-1994 (28) | / | 12.3 | 13910 |
| 9 | Northwest Branch Anacostia River (upper) | 1650500 | US | 1923-2013 (90) | 422 (330) | 18.3 | 5460 |
| 10 | South Branch Pennsauken Creek | 1467081 | US | 1967-2014 (47) | 203 (159) | 20.2 | 900 |
| 11 | Cooper River | 1467150 | US | 1963-2013 (50) | / | 21.1 | 4400 |
| 12 | Northeast Branch Anacostia River | 1649500 | US | 1938-2013 (75) | / | 21.7 | 18860 |
| 13 | Northwest Branch Anacostia River | 1651000 | US | 1938-2013 (54) | / | 22.7 | 12790 |
| 14 | Little Patuxent River | 1593500 | US | 1932-2013 (81) | / | 24.2 | 9840 |
| 15 | Stemmers Run | 1585300 | US | 1958-1989 (31) | 365 (365) | 28.5 | 1160 |
| 16 | Gwynns Falls | 1589300 | US | 1957-2013 (56) | 364 (273) | 30.0 | 8420 |
| 17 | Henson Creek | 1653500 | US | 1948-1978 (65) | / | 30.8 | 4330 |
| 18 | Watts Branch | 1645200 | US | 1957-1987 (30) | / | 35.3 | 960 |
| 19 | Whitemarsh Run | 1585100 | US | 1959-2013 (54) | 158 (92) | 36.2 | 1970 |
| 20 | West Branch Herring Run | 1585200 | US | 1957-2013 (56) | 579 (273) | 40.4 | 550 |
| 21 | East Branch Herbert Run | 1589100 | US | 1957-2013 (56) | 365 (273) | 44.4 | 640 |
| 22 | Dead Run | 1589330 | US | 1959-2013 (54) | 282 (189) | 48.6 | 1430 |
| 23 | Lang Lang River | 228209B | AUS | 1985 - 2011 (27) | 9 (2) | 0.0 | 27743 |
| 24 | McMahons Creek | 229106A | AUS | 1991 - 2008 (18) | 767 (128) | 0.0 | 4000 |
| 25 | OShannassy River | 229652A | AUS | 1979 – 2008 (30) | 1147 (196) | 0.0 | 8700 |
| 26 | Starvation Creek | 229109A | AUS | 1981 – 2008 (28) | 2071 (972) | 0.0 | 3160 |
| 27 | Olinda Creek | 229690 | AUS | 1992 - 2008 (17) | 302 (102) | 0.3 | 2610 |
| 28 | Woori Yallock Creek | 229679B | AUS | 2000 – 2011 (12) | 141 (31) | 3.0 | 28460 |
| 29 | Cardinia Creek | 228228A | AUS | 1985 - 2011 (27) | 262 (105) | 7.0 | 9060 |
| 30 | Bungalook Creek | 228369A | AUS | 1985 - 2011 (27) | 634 (236) | 13.0 | 385 |
| 31 | Cohranwarrabul Creek | 228393A | AUS | 1999 – 2011 (13) | 50 (24) | 20.0 | 6232 |
| 32 | Brushy Creek | 229249A | AUS | 1992 - 2008 (17) | 361 (281) | 28.0 | 1470 |
| 33 | Dandenong Creek @ Wantirna | 228357A | AUS | 1985 - 2011 (27) | 173 (94) | 31.0 | 6817 |
| 34 | Mullum Creek | 229648A | AUS | 1992 - 2008 (17) | 66 (13) | 31.0 | 3700 |
| 35 | Blind Creek @ Knox | 228366A | AUS | 1985 - 2011 (27) | 483 (281) | 41.0 | 1554 |
| 36 | Gardiners Creek @ Ashwood | 229625A | AUS | 1992 - 2008 (17) | 181 (72) | 44.0 | 3950 |
| 37 | Blind Creek @ High St | 228351A | AUS | 1985 - 2009 (25) | 758 (185) | 45.0 | 2656 |
| 38 | Gardiners Creek @ Kinkora | 229636A | AUS | 1985 - 2011 (27) | 1966 (213) | 47.0 | 432 |