# Peer review of "Technical note: Long-term persistence loss of urban streams as a metric for catchment classification"

_Hydrology and Earth System Sciences, 2017_

## Referee Comment (RC1) · A. Montanari (Referee) · 19 Nov 2017

The paper presents an interesting analysis where changes in correlations are used to measure human impact. A large set of catchment is considered to demonstrate the applicability of the above idea. The paper is excellently written and organized. I think it is presenting a relevant contribution.

The idea is very interesting and based on physical considerations. In fact, human impact affects the river flow regime by inducing changes in runoff formation. Urbanisation typically induces changes in travel time, as the flow formation is accelerated. Reduction of the travel time makes the hydrograph more peaky and less extended in time. These

changes imply corresponding variations in the autocorrelation function of the river flow time series, which can therefore be used to indirectly measure the human impact.

The paper uses the Hurst exponent as a measure of correlation. The idea is very interesting, but probably needs to be supported by some considerations on the physics of the underlying process. The use of the Hurst exponent implies the assumption that the underlying process is affected by long-term persistence (or long memory) which is not always present in river flows. Therefore, one may argue that the proposed metric is not efficient if long memory is not present in the original (not human impacted) process. To put the question in other words: the reader may wonder why the assumptions of long term persistent process was introduced for the unimpacted process (therefore using the H exponent as a measure of urbanization) instead of assuming that the original process is Markovian (therefore using, say, the lag-1 autocorrelation coefficient to measure urbanisation).

I am not against using the H exponent as a metric for measuring urbanisation, but I think a discussion should be provided for the its validity if long memory was not present in the underlying process. This discussion should take into account that long memory is an asymptotic behaviour and can therefore reliably measured only if long time series are available. When dealing with short series, estimation of the Hurst exponent is affected by large uncertainty and impacted by the presence of short memory (Markovian memory, which vanishes for increasing lags). Conversely, estimation of the lag-1 autocorrelation is affected by much less uncertainty.

Some of the series considered in this paper are very long, others may be too short for allowing a reliable estimation of the Hurst exponent. As a rule of thumbs, one may consider that it's difficult to identify long memory properties when the time series is not extended over several decades. Variability of the process, and the possible superimposition of a Markovian process over the long memory one, matter to determine uncertainty in long memory estimation.
I have a few minor remarks that the authors may consider when revising the paper.

1) Page 2, line 22: it is stated that "The Hurst exponent equals 0.5 for an uncorrelated white noise signal". This is not correct: the Hurst exponent equals 0.5 if the aggregated signal asymptotically converges to uncorrelated white noise. For instance, a Markovian process is correlated, but its Hurst exponent is 0.5. The reason is that a Markovian process asymptotically converges to white noise if aggregated.

2) Page 2, line 26: it is stated that "one would expect the Hurst exponent of urban streams to be closer to 0.5 when compared to rural and natural streams." This is the key of my reasoning above: what if the natural and urban stream has H=0.5, because the underlying process is, say, Markovian? Would the method be not applicable in these cases? In my opinion the idea would be applicable anyway if the lag-1 autocorrelation was considered as a metric instead of the Hurst exponent.

3) Page 3, line 10: the proposed deseasonalisation method works well for monthly data; when dealing with daily data, it leads to the estimation of seasonal averages and variances that are characterised by high day-to-day variability. If this is the case, they should be smoothed. It is possible that for the case of urban catchments the problem is less important. May be a short discussion could be provided. In any case, the problem is likely to be ineffective on long memory estimation.

4) Page 5, line 30: values of H around 0.60-0.65 may be hard to distinguish from 0.5, in view of the estimation bias and uncertainty.

5) Page 6, line 17: estimated values of H>1 have relevant implications on the nature of the underlying process that the authors should discuss. In my view they are likely to be due to estimation uncertainty. From a physical point of view, one should consider that H=1 is estimated for a non-stationary process like the Brownian motion, i.e., aggregation of a white noise. H=1.2 means that the process is non-stationary and, after differentiation, reduces to a stationary short memory process that is negatively correlated. Definitely, H=1.2 does not identify a stationary long memory process. It may be

that H=1.2 is herein obtained because the underlying process is non-stationary after urbanisation, but my feeling is that this is not the case.

6) In general the results are very interesting and definitely deserve to be published. My feeling is that the same results would be obtained by using the lag-one autocorrelation coefficient as metric, with the advantage that the method would rely on lighter assumptions and the underlying theory and practical application would be much easier.

As a final remark, I would like to point out that my preference for the lag-one autocorrelation as metric should be interpreted as my personal opinion, and not as a critic to this study. I am always in favor of simpler methods, but my opinion should be checked on the data and by no means should be taken as a suggestion to change the approach that has been taken here. I just would like to contribute to the discussion and to stimulate new ideas, but I am of course not sure that my intuition is correct. Definitely the overall idea that is presented by the authors is worth considering and the results deserve to be published.

Thank you for inviting me to review this paper.

Alberto Montanari

---

## Referee Comment (RC2) · P. S. Rao (Referee) · 28 Nov 2017

1
 Review Comments Technical Note MS# hess-2017-613 MS Type

 2
 P. Suresh C. Rao and Leonardo E. Bertassello, Lyles School of Civil Engineering

 3
 Purdue University, West Lafayette, IN 47907, USA (SureshRao@purdue.edu)

 4
 [Long-term memory loss of urban streams as a metric for catchment classification by

 5
 Dusan Jovanovic, Tijana Jovanovic, Alfonso Mejía, Jon Hathaway, and Edoardo Daly]

6

7 Jovanovic et al (2017) present a case study for the use of Hurst exponent to evaluate the 8 impacts of increasing urbanization on stream hydrological responses. This approach represents 9 an alternative way to analyze the long-term correlation between rainfall and streamflow time series. Increasing urbanization (e.g., impervious area; engineered drainage networks) contributes 10 to increasing "flashiness" of stream flow, with loss of landscape "buffering" through infiltration, 11 ET losses, and slow recession. Previous studies (Yang et al. 2010) have shown that for impervious 12 13 surface area (ISA) between 5-35%, a linear increase in frequency of high-flow events; beyond this range, urbanization impacts on stream hydrologic regime are expected to be nonlinear. In a 14 hypothetical case of 100% impervious area and highly connected urban drainage infrastructure 15 network, rainfall events are quickly translated to stream discharge, assuming minimal storage. 16 Thus, time series of rainfall and discharge are highly correlated, especially for the larger events. 17 18 Yang and Bowling (2014) examined changes in hydrologic system memory for sixteen basins with 19 varying degrees of urbanization in the Great Lakes region. They concluded that decrease in longterm memory in simulated streamflow with increasing urbanization relates to a decreased low-20 frequency power and amplitude of soil-water storage. Kim et al [2015] used power spectral 21 22 analyses for several urbanizing watersheds in South Korea, to show that slopes of power spectra for discharge time series converge to that of rainfall with increasing urbanization, a clear evidence for loss of "memory" or "landscape buffering". In un-impacted streams, discharge time series is characterized as  $1/f^{\alpha}$  noise, with  $\alpha \sim 1$  (Godsey et al., 2015).

26 In the analyses Jovanovic et al (2017) presented here, Hurst exponent (H) approaches 0.5 for time-series of uncorrelated, independent, random variable; this is usually the case of rainfall 27 aggregated at daily scale (white-noise signal) with stationary patterns. Larger H values are related 28 to long-term correlation (memory, persistency), that are typical of discharge in non-urbanized 29 30 streams. It is then expected that with increasing urbanization, H for urban stream flows would 31 shift towards H for rainfall, as illustrated in Figures 1 and 2 of Jovanovic et al (2017) for urban watersheds with impervious area up to ~50%. It would be interesting to examine other urbanized 32 33 watersheds with imperviousness are much larger than 50%, as was the case for Kim et al (2015). Figure 3 in Jovanovic et al (2017) reveals weak correlations between H value for stream discharge 34 and catchment size, annual rainfall, and area-normalized mean discharge. Thus, the dominant 35 36 control on dampening of rainfall time series – introducing "memory" -- is landscape storage and loss dynamics. 37

Hurst exponent for rainfall time series would allow comparisons between *H* values for natural, peri-urban and urban catchments. That is, does urbanization not only impact the stream flow but also rainfall patterns over the urbanized area, relative to the non-urbanized or periurban areas? Furthermore, non-stationarity of rainfall patterns (e.g., seasonality; or long-term shifts) will also result in  $H \neq 0.5$ . It is well known that urbanization modifies local atmospheric conditions enough to alter rainfall patterns and total amounts (Niyogi et al., 2017; Sheng et al., 20??). Thus, the rainfall *H* values might be different for the urban, peri-urban and rural areas.

**46 Literature Cited**

47 Godsey, S., Aas, W, Clair, T., deWit, H., Fernandez, I, Kahl, S., Malcolm, I., Neal, C., Neal, 48 M., Nelson, S., Norton, S., Palucis, M., Skelkavale, B., Soulsby, C., Terzalff, D., and Kirschner, J. 49 Generality of fractal 1/f scaling in catchment tracer time series, and its implications for 50 51 catchment travel time distributions. *Hydrol. Processes*, 24(12): 1660-1671. 52 53 Kim, D.H., Rao, P.S.C., Kim, D., and Park, J. 2015. 1/ f noise analyses of urbanization 54 effects on streamflow characteristics, Hydrol. Processes, 30(11):1651-1664. 55 DOI: 10.1002/hyp.10727. 111 112 Niyogi, D., Lei, M., Kishtawi, C., Schmid, P. 2017. Urbanization Impacts on the Summer Heavy Rainfall Climatology over the Eastern United States, Earth Interactions, June 2017, 113 114 https://doi.org/10.1175/EI-D-15-0045.1 115 116 Sheng, C., Li, W., Du, Y, Mao, C., and Zhang, L. Urbanization effect on precipitation over the Pearl River Delta based on CMORPH data. Advances in Climate Change Research, 6(1): 16-22. 117 118 119 Yang, G., and L. C. Bowling, 2014. Detection of changes in hydrologic system memory associated with urbanization in the Great Lakes region, Water Resour. Res., 50, 3750–3763, 120 121 doi:10.1002/2014WR015339. 122 123 Yang, G., Bowling, L.C., Cherkauer. K.A., Pijanowski, B.C., Niyogi, D. 2010. Hydroclimatic 124 response of watersheds to urban intensity: An observational and modeling-based analysis for 125 the White River Basin, Indiana. Jour. Hydrometeor. 11:122-138. DOI: 10.1175/2009JHM1143.1

---

## Referee Comment (RC3) · P. Dimitriadis (Referee) · 30 Nov 2017

General comments

The paper further investigates the effect of urbanization on stream flows through the long-term (i.e. in large scales or lags) change in the process second-order dependence structure (e.g. autocorrelation function). The review of previous studies is presented in the introduction [section 1] but I believe some additional literature review is required (please see in the minor comments). The above quantification is accomplished for (a) the urbanization through the measurement of the catchment imperviousness (in particular, they suggest 3 ranges of urbanization, i.e. natural, peri-urban and urban,

that correspond to less than 5% imperviousness, in between 5% and 15%, and larger than 15%, respectively) [sections 1 and 3], and (b) the long-term alteration through the estimation of the Hurst parameter (applying two statistical methods, i.e. R/S and MF-DFA) [sections 2.1 and 3]. The proposed methodology is applied to 38 catchments (22 in USA and 16 in Australia, some with no missing data and highlighting the results from the ones with missing data) [sections 2.2 and 3]. Furthermore, the Authors calculate the statistical significance between catchment size, annual rainfall and specific mean discharge (i.e. discharge over catchment size), with no clear significance found (the Authors mention that a weak significance is found between H and annual rainfall as well as specific mean discharge attributing it to the small catchment sizes used) [section 3]. The main conclusion is that a correlation is evident between catchment imperviousness and the Hurst parameter and so, the Authors suggest using the latter as an index/metric of the former [sections 3 and 4].

In my opinion, the idea (of adding H to the several urbanization metrics) looks promising and certainly is worth of attention. The paper is well written and well structured, and previous studies are well documented. The generalization of the results is justified since the proposed methodology has been applied to several catchments in different climatic conditions and continents. However, I believe that the analysis still needs to be improved. Below, I have numbered several suggestions and comments that I hope the Authors will find useful for their analysis and worth of discussing.

Major comments:

1) The basic idea of linking catchment urbanization to the long-term alteration of stream flows is based on the assumption that both are well (and independently) quantified. However, the urbanization is quantified through the catchment imperviousness which is certainly an effect of urbanization but not exclusively, and thus, it may be useful to include additional metrics for the classification of the catchments (e.g. land-use). If (after including other metrics) the classification presented by the Authors still stands then this could be an additional finding of the paper, i.e. that urbanization can be well

classified just by using the imperviousness metric.

An additional possible limitation of just using the imperviousness metric for the urbanization classification, is that there may be catchments that have a small imperviousness metric (and thus, they must have been classified here as natural or peri-urban) but may include upstream civil works (such as dams) which certainly have an effect on the Hurst parameter and in particular, these upstream constructions are expected to cause a large drop of H close to 0.5 (since the release of water in the river would be no longer entirely dependent to the rainfall-runoff natural process). It may be useful for the Authors to check whether such large-scale civil works exist upstream of the stream flow stations, and if this is the case, consider adding a discussion (or even introduce a new classification for them).

Finally, since both imperviousness and streamflow H are affected by urbanization in various ways (and so, a direct comparison between them may not be always illustrative of the urbanization level), the Authors may consider to additionally estimate the H of precipitation at the examined locations of the streamflow stations or to nearby stations within the catchment. If the decreasing level of streamflow H to imperviousness is repeated for the precipitation H (as shown in Figure 1 of the paper), then this will strengthen the robustness of the analysis (based also on the results of Jovanovic et al., 2016 mentioned in the paper). An investigation of the precipitation over the examined catchments is also suggested by a Referee of this paper.

2) The H parameter is one of the key factors of the analysis, and although the Authors have used two methods to estimate it, both suggested methods do not take into account the bias effect (e.g. Tyralis and Koutsoyiannis, 2011) which can be very large for long-term persistent processes and especially when estimated from short-length time series (e.g. Koutsoyiannis, 2013).

Also, Dimitriadis and Koutsoyiannis (2015a) have shown (through Monte-Carlo analysis of a wide range of long-term persistent processes) that one requires at least the

10% of a time series length to estimate the Hurst parameter adjusted for bias, which corresponds to at least 20 years of measurements (the Authors mention this in Ln. 263 but with no references and so, they may consider using this reference to justify their statement), so as to have at least 2 values to estimate the log-log slope of the dependence structure and thus, the H parameter adjusted for bias. Note that the adjustment for bias usually increases the H (< 1) estimation and thus, it could be also mentioned in the paper that.

The above comment on bias is also mentioned from another Referee of this paper. Also, this Referee suggests using the lag-1 autocorrelation coefficient instead of the H parameter. This could be easily done by the Authors if they decide to follow this suggestion and then, they could see how the lag-1 coefficient is linked to the Hurst parameter (normally higher coefficients will correspond to higher H) they have already estimated. I believe this could strengthen the robustness of the analysis even more (but again I believe the effect of bias to the lag-1 coefficient should again be mentioned).

An additional issue worth of discussing is the estimations of H > 1. The Hurst parameter corresponds to the large lag (or scale) behaviour of an ergodic (and thus, stationary) process (Koutsoyiannis and Montanari, 2015) like for example the fractional Gaussian noise, and can be easily quantified (without adjusting for bias) through the log-log slope of e.g. the autocorrelation function. Therefore, an H > 1, corresponds to an increasing autocorrelation function with lag, which comes into contradiction with the originally assumption of ergodicity. As already mentioned by another Referee of this paper, I also believe the H > 1 estimated values in the paper are due to sampling errors and not to non-stationarity (which as explained above the latter conclusion leads to a contradiction).

3) In the Abstract, it is mentioned that '. . .the relationship between this exponent and level of urbanisation needs to be further examined and verified on catchments with different levels of imperviousness and from different climatic regions' [Ln. 14-15]. However, I could not find in the analysis the effect of the (properly defined) climatic conditions of the catchments to the H parameter. If the Authors would like to add this to the analysis, they could easily do so. Since both temperature and precipitation is already included in the analysis [section 2.2] as well as comments on some climatic impacts [end of section 3], I think it would be useful for the Readers and in favor of the generalization of the analysis, to assign a climatic regime metric (e.g. just the five basic classifications of Koppen-Geiger; http://koeppen-geiger.vu-wien.ac.at/) to each catchment and add a discussion about how (or whether) this has an effect on the H of streamflows and precipitations (it is my belief that different Koppen-Geiger classifications will have different H, .e.g. Markonis et al., 2016; Tyralis et al., 2017).

Minor comments:

1) In my opinion that additional references should be included in the Introduction. Some of previous works that the Authors may find interesting are O'Driscoll et al. (2010), Miller et al. (2014), and references therein.

2) In my opinion, the equations in section 2.1 describing the two methods should be placed in an Appendix.

3) In Ln. 83-84 the Authors mention that the "Seasonal cycles are removed from the original series by subtracting the calendar day mean and dividing by the calendar day standard deviation". However, this is true only for a Gaussian process (streamflows are not-Gaussian distributed). In my opinion the word 'removed' should be replaced with 'approximately removed' or apply a more robust method for de-seasonalization (e.g. Dimitriadis and Koutsoyiannis, 2015b), where each cycle is modeled through a time-varying parameter of the distribution function (which may not be necessarily Gaussian).

4) Please, consider adding extra information on the data used in the analysis, as for example climatic regime (e.g., through the Koppen-Geiger classification), latitude and longitude, percentage of zero values (if any) as well as some statistical characteristics such as mean, standard deviation etc.

5) Ln. 176: how is imperviousness U parameter mathematically defined (for example does it take into account the land-use). I think that it would be easier for the Readers to understand the proposed analysis if a mathematical expression of this parameter is included.

6) In Figure 1 please add somewhere in the legend the process name, i.e. 'stream-flows'.

7) Ln. 211-213. The Authors mention that "Between 5% and 15% the values of H appear more scattered. Therefore, the three levels of imperviousness defined from Figure 1 can be classified as natural (U<=5%), peri-urban (5%15%) catchments based on the corresponding range of values of H.". In my opinion, it is not clear how the natural and peri-urban classification is justified. Perhaps the Authors could add some comments on this and make use of other rather simple statistical metrics to define the limit of 5% between natural and peri-urban catchments. An example could be to justify this value (of 5%) of imperviousness where H is estimated larger than 0.75 (for the R/S method), which is the mean between 0.5 (white noise behaviour at large scales) and $\sim$1 (highly persistent).

8) Ln. 196-197: "This suggests that the increase in impervious cover might cause more precipitation to bypass the groundwater storage", and Ln. 232-233: "Therefore, small catchments may not have sufficient water storage to influence the long-term dependence in flow time series.".

The above sentences seem somehow contradictory, in the sense that if there is no clear relationship between H and catchment size because the catchment sizes are small compared to literature, then how come H is decreasing due to the increase of the imperviousness which has caused more precipitation to bypass the storage capacity? I believe the Authors try to explain this in Ln. 237-239 by mentioning that "Generally, catchments with lower rainfall totals and lower specific mean streamflow are found to have higher long-term dependence due to the longer dry weather periods and con-

sequently longer low flow periods.". However, only minor information is given for the rainfall totals and aridity of the examined catchments. Maybe some further explanation here (and more information provided on the climatic conditions of the examined catchments) could help the Readers to better understand this point.

Spelling and grammar comments:

1) Ln. 187: "15% imperviousness". Please replace this with '15% of imperviousness'. 2) Ln. 192: "... due the high precipitation...". Please replace 'due the high' with 'due to high'.

I hope the Authors would find some of the above comments useful to their analysis.

Thank you for the invitation to review this paper.

Panayiotis Dimitriadis

References

Dimitriadis, P., and D. Koutsoyiannis, Climacogram versus autocovariance and power spectrum in stochastic modelling for Markovian and Hurst–Kolmogorov processes, Stochastic Environmental Research & Risk Assessment, 29 (6), 1649–1669, doi:10.1007/s00477-015-1023-7, 2015a.

Dimitriadis, P., and D. Koutsoyiannis, Application of stochastic methods to double cyclo-stationary processes for hourly wind speed simulation, Energy Procedia, 76, 406–411, doi:10.1016/j.egypro.2015.07.851, 2015b.

Jovanovic, T., Mejia, A., Gall, H. & Gironas, J., Effect of urbanization on the long-term persistence of streamflow 295 records, Physica A-Statistical Mechanics and Its Applications, 447, 208-221, 2016.

Koutsoyiannis, D., Hydrology and Change, Hydrological Sciences Journal, 58 (6), 1177–1197, doi:10.1080/02626667.2013.804626, 2013.

Koutsoyiannis, D., and A. Montanari, Negligent killing of scientific concepts: the stationarity case, Hydrological Sciences Journal, 60 (7-8), 1174–1183, doi:10.1080/02626667.2014.959959, 2015.

O'Driscoll, M., S. Clinton, A. Jefferson, A. Manda, and S. McMillan. 2010. Urbanization effects on watershed hydrology and in-stream processes in the southern United States, Water, 2(3):605–648, doi:10.3390/w2030605.

Markonis, Y., C. Nasika, Y. Moustakis, A. Markopoulos, P. Dimitriadis, and D. Koutsoyiannis, Global investigation of Hurst-Kolmogorov behaviour in river runoff, European Geosciences Union General Assembly 2016, Vol. 18, EGU2016-17491, doi:10.13140/RG.2.2.16331.59684, European Geosciences Union, 2016.

Miller, J.D., H. Kim, T.R. Kjeldsen, J. Packman, S. Grebby, R. Dearden, Assessing the impact of urbanization on storm runoff in a peri-urban catchment using historical change in impervious cover, Journal of Hydrology, 515 (2014) 59–70, 2014.

Tyralis H., and D. Koutsoyiannis, Simultaneous estimation of the parameters of the Hurst-Kolmogorov stochastic process, Stochastic Environmental Research & Risk Assessment, 25 (1), 21–33, 2011.

Tyralis, H., Dimitriadis, P., Koutsoyiannis, D., O'Connell, P.E., Tzouka, K. and Iliopoulou, T., On the long-range dependence properties of annual precipitation using a global network of instrumental measurements, Advances in Water Resources, 2017.

---

## Author Comment (AC1) · 12 Jan 2018

We thank Dr Montanari for the thorough review of the manuscript, and the encouraging comments and suggestions for its improvement. Two main points were raised by Dr Montanari: 1) The use of the H-exponent as a metric to determine the role of urbanization on streamflow implies the assumption that all natural catchments show long term memory. This might be not true and, perhaps, the lag-1 autocorrelation coefficient could be a better candidate for the same analysis. 2) The values of the H-exponent larger than 1 need to be interpreted and explained. Very likely, they are due to estimation uncertainties. The discussion of this point required to be strengthen in the

manuscript. In relation to the first issue, we will expand the discussion about the use of H exponent and will repeat the analysis with the lag-1 autocorrelation coefficient to test the hypothesis suggested. If these new results provide substantial contributions to the study, we will expand the manuscript to include also the lag-1 autocorrelation coefficient in the analysis. This might also help with the series having long gaps of consecutive days. For the second point, we will explore the causes leading to values of H larger than 1 obtained with the MF-DFA. We agree with Dr Montanari that these could be caused to estimation uncertainties: a discussion on this issue will be included in the revised manuscript. All minor comments related to simple corrections of the text will be addressed in the revised manuscript. Minor comment 3) criticizes the deseasonalisation method we used. Because this issue was already raised during the review process of the paper by Jovanovic et al. (2017), they compared several deseasonalisation methods for the same 22 streamflow time series in the USA; this led to very similar results. Therefore, due to previous findings and the scope of this technical note, we will not perform any additional analysis, and provide a stronger justification of the deseasonalisation method used.

References: Jovanovic, T., García, S., Gall, H. and Mejía, A., 2017. Complexity as a streamflow metric of hydrologic alteration. Stochastic Environmental Research and Risk Assessment, 31(8), pp.2107-2119.

---

## Author Comment (AC2) · 12 Jan 2018

We thank Dr Rao and Mr Bertassello for reviewing our manuscript and providing useful comments. The review focuses more on rainfall and the role of rainfall statistics in relation to urbanization. In highly urbanised catchments, rainfall is rapidly transferred to streams, thereby suggesting that the H-exponents of rainfall and streamflow time series should have values close to each other. Conversely, the ability of more natural catchments to store and lose water while slowly releasing it to streams should introduce a longer memory on the streamflow time series when compared to rainfall. This is in agreement with the results already presented in Jovanovic et al. (2016), who used

the same US catchments of the present study to show that the scaling properties of quickflow in these station were similar to those of some rainfall stations in the same area. In the present study, although an interesting research idea, we cannot provide a full analysis of rainfall and streamflow statistics, because we do not have enough rainfall stations in the USA (in addition to those already used in Jovanovic et al. (2016)) and Australia to compare with the streamflow stations. Furthermore, the main focus of our technical note is to show that catchments can be classified in different categories of urbanization levels solely by using the H-exponent of streamflow series. This exponent can be estimated in relatively simple ways from streamflow series, which are becoming increasingly more available from water management agencies. We used streamflow series from the USA and Australia to show that a relationship between the H-exponent of streamflow series and the fraction of catchment impervious area could be found in parts of the world with different climatic conditions. The issue of urbanization altering rainfall patterns is interesting but outside the scope of our technical note. We will expand the sections with discussion and conclusions to better explain the link between streamflow and rainfall, and to provide the suggestion of analysing the link between urbanization levels and rainfall.

References: Jovanovic, T., Mejia, A., Gall, H. & Gironas, J. 2016. Effect of urbanization on the long-term persistence of streamflow records. Physica a-Statistical Mechanics and Its Applications, 447, 208-221.

---

## Author Comment (AC3) · 12 Jan 2018

We thank Dr Dimitriadis for the very thorough review and the suggestions, which will help largely improve the manuscript. We have summarised below the major comments that were raised and have indicated how we plan to address them.

1) The urbanization is quantified through the catchment imperviousness which is certainly an effect of urbanization but not exclusively, and thus, it may be useful to include additional metrics for the classification of the catchments (e.g. land-use).

For the Australian catchments, the percentage of impervious area were taken from

Hamel et al. (2015) and were calculated using the methods in Kunapo et al. (2005). Buildings and paved areas (i.e., roads and carparks) were mapped using a geographical information system software, and the percentage of impervious areas, U, was calculated as the ratio between total impervious area and catchment area. The land use of the catchments with larger U is residential, while those with low U are mostly covered by natural forests. For the US catchments, information about impervious data can be found in Mejia et al. (2015). Imperviousness was calculated using a combination of tax map information and areal imagery. In terms of land use, pervious land is mainly agricultural and urban green spaces, including lawns, parks and other grassed areas. Riparian corridors, if present, are likely to be forested. The majority of imperviousness consist of residential, commercial and transportation land use areas.

Existence of catchments with small imperviousness metric, but having civil works (e.g. dams) which can influence the H exponent.

The catchments selected were not affected by large civil works such as dams. We will specify this in the text.

Investigate precipitation and imperviousness correlation to see if the same pattern will emerge with precipitation which will strengthen the robustness of the analysis.

As specified in the reply to Dr Rao and Mr Bertassello, the links between rainfall and streamflow in catchments with different levels or urbanization were analysed by Jovanovic et al. (2016), who found that the scaling properties of streamflow records for highly urbanized catchments were similar to those of the rainfall records. The focus of the present technical note is to show that a relationship between the H-exponent of streamflow series and percentage of impervious area can be found in different climatic conditions (i.e., humid climate for the catchment in the USA and semi-arid for those in Australia). This relationship identifies levels of urbanization, defined in terms of percentage of impervious area of a catchment, U, that are associated with different degrees of long-term memory of streamflow series, estimated using the H-exponent.

We will expand the discussion to direct readers to Jovanovic et al. (2016) in relation to the H-exponent of rainfall series in catchment with different degrees of urbanization.

2) Examine the lag-1 correlation function relationship to H exponent, which may strengthen the robustness of the analysis.

As replied to Dr Montanari, we will conduct the lag-1 analysis and possibly include the results in the manuscript.

Issue of H>1.

This issue, which was also raised by Dr Montanari, will be addressed by expanding the discussion and analysing the uncertainty of the H-exponent estimation.

3) Assign a climatic regime metric (e.g. just the five basic classifications of Koppen-Geiger; http://koeppen-geiger.vu-wien.ac.at/) to each catchment and add a discussion about how (or whether) this has an effect on the H of streamflows and precipitations.

We will insert the Koppen-Geiger classification for our catchments and provide a discussion about the impact of different climates on H-exponent estimation.

All minor comments will be addressed in the manuscript and the suggested references will be appropriately included in the revised manuscript.

References:

Jovanovic, T., Mejia, A., Gall, H. & Gironas, J. 2016. Effect of urbanization on the long-term persistence of streamflow records. Physica a-Statistical Mechanics and Its Applications, 447, 208-221.

Kunapo, J., Sim, P.T. & Chandra, S., 2005. Towards automation of impervious surface mapping using high resolution orthophoto. Applied GIS, 1 (1), 03.1-03.19.

Mejía A, Rossel F, Gironás J, & Jovanovic T 2015. Anthropogenic controls from urban growth on flow regimes. Adv Water Resour 84:125–135.

doi:10.1016/j.advwatres.2015.08.010

---

## Author Response (AR1)

**Response to Reviewers' Comments**

**Article title: Technical note: Long-term memory loss of urban streams as a metric for**

**catchment classification**

Reference No: HESS-2017-613

**Reviewer #1 – A. Montanari**

The paper presents an interesting analysis where changes in correlations are used to measure human impact. A large set of catchment is considered to demonstrate the applicability of the above idea. The paper is excellently written and organized. I think it is presenting a relevant contribution.

The idea is very interesting and based on physical considerations. In fact, human impact affects the river flow regime by inducing changes in runoff formation. Urbanisation typically induces changes in travel time, as the flow formation is accelerated. Reduction of the travel time makes the hydrograph more peaky and less extended in time. These changes imply corresponding variations in the autocorrelation function of the river flow time series, which can therefore be used to indirectly measure the human impact.

We thank Dr Montanari for the thorough review of the manuscript, and the encouraging comments and suggestions for its improvement.

The paper uses the Hurst exponent as a measure of correlation. The idea is very interesting, but probably needs to be supported by some considerations on the physics of the underlying process. The use of the Hurst exponent implies the assumption that the underlying process is affected by long-term persistence (or long memory) which is not always present in river flows. Therefore, one may argue that the proposed metric is not efficient if long memory is not present in the original (not human impacted) process. To put the question in other words: the reader may wonder why the assumptions of long term persistent process was introduced for the unimpacted process (therefore using the H exponent as a measure of urbanization) instead of assuming that the original process is Markovian (therefore using, say, the lag-1 autocorrelation coefficient to measure urbanisation).

I am not against using the H exponent as a metric for measuring urbanisation, but I think a discussion should be provided for its validity if long memory was not present in the underlying process. This discussion should take into account that long memory is an asymptotic behaviour and can therefore be reliably measured only if long time series are available. When dealing with short series, estimation of the Hurst exponent is affected by large uncertainty and impacted by the presence of short memory (Markovian memory, which vanishes for increasing lags). Conversely, estimation of the lag-1 autocorrelation is affected by much less uncertainty.

Some of the series considered in this paper are very long, others may be too short for allowing a reliable estimation of the Hurst exponent. As a rule of thumbs, one may consider that it's difficult to identify long memory properties when the time series is not extended over several decades. Variability of the process, and the possible superimposition of a Markovian process over the long memory one, matter to determine uncertainty in long memory estimation.

We thank Dr Montanari for raising these points. Accordingly, some modifications were made to the manuscript:

- The title was slightly changed and the term 'persistence' is now adopted instead of 'memory', which might be misleading. The term memory throughout the manuscript was also changed into persistence.
- Although it is true the long-term persistence of non-urbanized catchments was taken as an assumption, this assumption appears to be correct in the case of the catchments that we analyzed. We made this clearer in the text, by adding at page 2 of the annotated manuscript (lines 33-35):

"The assumption that catchments with lower degrees of urbanization present long-term persistence needs also to be validated across a spectrum of catchments."

- We calculated the lag-1 autocorrelation coefficients of the de-seasonalized series of our catchments and show this in Figures 1 and 3. Text was added accordingly in different parts of the manuscript (refer to annotated manuscript).

Page 5, line15: "Given the uncertainty in the estimation of H, especially when streamflow series are shorter than twenty years, the Pearson autocorrelation function of the series was calculated and the autocorrelation coefficient at 1-day delay (i.e., lag-1 Pearson autocorrelation) was selected as a metric characterizing the persistence of the series. The values of the lag-1 autocorrelation coefficient where then related to levels of catchment imperviousness as done for the H exponent."

Page 6, line 28: "The lag-1 autocorrelation coefficient follows a pattern similar to H for the Australian catchments, but the relationship between the lag-1 autocorrelation coefficient and imperviousness is not as evident for the catchments in the USA (Figure 1). The US catchments can be quite flashy, and this may be a reason why the lag-1 autocorrelation is low irrespectively of the level of imperviousness."

Page 7, line 20: "A similar classification would result from the lag-1 autocorrelation coefficient for the Australian catchments, while the lag-1 autocorrelation would not show any visible difference in the US catchments.

Because other variables and catchment attributes can affect the value of H, the estimated H exponents and lag-1 autocorrelation coefficients were also related to the catchment area and catchment wetness.

The differences in lag-1 autocorrelation coefficients between US and Australian catchments shown in Figure 1 make it difficult to identify a general pattern in relation to other variables (Figure 3)."

I have a few minor remarks that the authors may consider when revising the paper.

1) Page 2, line 22: it is stated that "The Hurst exponent equals 0.5 for an uncorrelated white noise signal". This is not correct: the Hurst exponent equals 0.5 if the aggregated signal asymptotically converges to uncorrelated white noise. For instance, a Markovian process is correlated, but its Hurst exponent is 0.5. The reason is that a Markovian process asymptotically converges to white noise if aggregated.

We have corrected this statement in the revised manuscript. The phrase now reads (page 2 of the annotated manuscript, line 23):

"The Hurst exponent tends to 0.5 when an aggregated signal converges to white noise, while values larger than 0.5, as commonly found for streamflow series, are associated with persistent processes (e.g., Serinaldi, 2010)."

2) Page 2, line 26: it is stated that "one would expect the Hurst exponent of urban streams to be closer to 0.5 when compared to rural and natural streams." This is the key of my reasoning above: what if the natural and urban stream has H=0.5, because the underlying process is, say, Markovian? Would the method be not applicable in these cases? In my opinion the idea would be applicable anyway if the lag-1 autocorrelation was considered as a metric instead of the Hurst exponent.

See replies to the comments above about the lag-1 autocorrelation.

3) Page 3, line 10: the proposed deseasonalisation method works well for monthly data; when dealing with daily data, it leads to the estimation of seasonal averages and variances that are characterised by high day-to-day variability. If this is the case, they should be smoothed. It is possible that for the case of urban catchments the problem is less important. May be a short discussion could be provided. In any case, the problem is likely to be ineffective on long memory estimation.

With similar data used for another study, we have tried different deseasonalisation procedures, such as ensemble empirical mode decomposition (EEMD), which allows smoothing the variability. We found that this only had a marginal effect on the results. Moreover, the day-to-day variability in urban streams might be related to the imperviousness of the catchment, and smoothing the series might artificially remove differences between catchments with different levels of urbanization. Thus, we think the approach we used to approximately remove the seasonal cycles is reasonable.

4) Page 5, line 30: values of H around 0.60-0.65 may be hard to distinguish from 0.5, in view of the estimation bias and uncertainty.

We agree with this statement; we report here the values that were estimated with the two methods (i.e., R/S and MF-DFA) to show that, although estimates higher than 0.5 are obtained, these are anyway lower than the values of H estimated for catchments with lower degrees of imperviousness.

5) Page 6, line 17: estimated values of H>1 have relevant implications on the nature of the underlying process that the authors should discuss. In my view they are likely to be due to estimation uncertainty. From a physical point of view, one should consider that H=1 is estimated for a non-stationary process like the Brownian motion, i.e., aggregation of a white noise. H=1.2 means that the process is nonstationary and, after differentiation, reduces to a stationary short memory process that is negatively correlated. Definitely, H=1.2 does not identify a stationary long memory process. It may be that H=1.2 is herein obtained because the underlying process is non-stationary after urbanisation, but my feeling is that this is not the case.

We modified the text to avoid misunderstanding on the values of H>1. At page 6 of the annotated manuscript (line 15), we now write: "The values of *H* obtained using R/S analysis were between about 0.6 to around 0.9, while the MF-DFA analysis resulted in slightly higher values, with H larger than 0.65 (Figure 1); MF-DFA can give estimates slightly higher than 1 due to uncertainties in the estimation of the parameter H (Serinaldi, 2010)."

At page 7 (line 6), we now write: "Catchments with less than 5% impervious areas have larger values of H (0.75

We agree with Dr Dimitriadis. We have now modified the text accordingly. The first phrase of the results (page 6 of the annotated manuscript, line 15) now reads:

"The values of *H* obtained using R/S analysis were between about 0.6 to around 0.9, while the MF-DFA analysis resulted in slightly higher values, with H larger than 0.65 (Figure 1); MF-DFA can give estimates slightly higher than 1 (Serinaldi, 2010) due to uncertainties in the estimation of the parameter H."

In the Abstract, it is mentioned that '...the relationship between this exponent and level of urbanisation needs to be further examined and verified on catchments with different levels of imperviousness and from different climatic regions' [Ln. 14-15]. However, I could not find in the analysis the effect of the (properly defined) climatic conditions of the catchments to the H parameter. If the Authors would like to add this to the analysis, they could easily do so. Since both temperature and precipitation is already included in the analysis [section 2.2] as well as comments on some

climatic impacts [end of section 3], I think it would be useful for the Readers and in favor of the generalization of the analysis, to assign a climatic regime metric (e.g. just the five basic classifications of Koppen-Geiger; http://koeppen-geiger.vuwien.ac.at/) to each catchment and add a discussion about how (or whether) this has an effect on the H of streamflows and precipitations (it is my belief that different Koppen-Geiger classifications will have different H, .e.g. Markonis et al., 2016; Tyralis et al., 2017).

We have added at page 5 of the annotated manuscript (line 36):

"According to the Koppen-Geiger classification, the climate of the US and Australian catchments is classified as Cfa and Cfb respectively."

**Minor comments:**

1) In my opinion that additional references should be included in the Introduction. Some of previous works that the Authors may find interesting are O'Driscoll et al. (2010), Miller et al. (2014), and references therein.

Following this suggestion, we added a reference to O'Driscoll et al. (2010), who provide a general overview of the effect of urbanization of streamflow. Miller et al. (2014) appears to be more a case study and we prefer not to add it; we added Fletcher et al. (2013), which is a more general review. Considering that this is a technical note, we believe that the background provided in our introduction is enough to direct readers to the literature required to put the study in the appropriate context.

2) In my opinion, the equations in section 2.1 describing the two methods should be placed in an Appendix.

Because this is a technical note, we would prefer to avoid the use of an Appendix and leave the brief description of the methods in the main text.

3) In Ln. 83-84 the Authors mention that the "Seasonal cycles are removed from the original series by subtracting the calendar day mean and dividing by the calendar day standard deviation". However, this is true only for a Gaussian process (streamflows are not-Gaussian distributed). In my opinion the word 'removed' should be replaced with 'approximately removed' or apply a more robust method for de-seasonalization (e.g. Dimitriadis and Koutsoyiannis, 2015b), where each cycle is modeled through a time varying parameter of the distribution function (which may not be necessarily Gaussian).

We added "approximately removed". We tried in the past, for another study, using a more complex method to remove seasonal cycles, namely ensemble empirical mode decomposition (EEMD), and found that it only had a marginal effect on the results. Also, a similar conclusion was found before using MF-DFA (see, Livina et al. 2011, Chapter 13: Seasonality Effects on Nonlinear Properties of Hydrometeorological Records, in *In Extremis*, Springer Berlin Heidelberg, pp. 266-284), where they compared the approach used in this study against the so called phase-substitution approach, which is based on the Fourier phases of the time series.

4) Please, consider adding extra information on the data used in the analysis, as for example climatic regime (e.g., through the Koppen-Geiger classification), latitude and longitude, percentage of zero values (if any) as well as some statistical characteristics such as mean, standard deviation etc.

Ranges of annual precipitation and temperatures were already provided. The climate classification has been added in the revised manuscript. At page 5 of the annotated manuscript (line 36), we write:

"According to the Koppen-Geiger classification, the climate of the US and Australian catchments is classified as Cfa and Cfb respectively."

5) Ln. 176: how is imperviousness U parameter mathematically defined (for example does it take into account the land-use). I think that it would be easier for the Readers to understand the proposed analysis if a mathematical expression of this parameter is included.

A more detailed description of the calculation of U is provided at the end of Section 2.2 (page 6 of the annotated manuscript, line 3). This reads:

"For the US catchments, information about impervious data can be found in Mejia et al. (2015). Imperviousness was calculated using a combination of tax map information and areal imagery. In terms of land use, pervious land is mainly agricultural and urban green spaces, including lawns, parks and other grassed areas. Riparian corridors, if present, are likely to be forested. The majority of imperviousness consist of residential, commercial and transportation land use areas. For the Australian catchments, the percentage of impervious area were taken from Hamel et al. (2015) and were calculated using the methods in Kunapo et al. (2005). Buildings and paved areas (i.e., roads and carparks) were mapped using a geographical information system software, and the percentage of impervious areas, U, was calculated as the ratio between total impervious area and catchment area. The land use of the catchments with larger U is residential, while those with low U are mostly covered by natural forests."

6) In Figure 1 please add somewhere in the legend the process name, i.e. 'streamflows'.

In the caption of the figures we now specified that the H-exponent refer to the streamflow series. The figure has been revised to add the lag-1 autocorrelation coefficient.

7) Ln. 211-213. The Authors mention that "Between 5% and 15% the values of H appear more scattered. Therefore, the three levels of imperviousness defined from Figure 1 can be classified as natural (U <= 5%), peri-urban (5% < U <= 15%), and urban (U > 15%) catchments based on the corresponding range of values of H.". In my opinion, it is not clear how the natural and peri-urban classification is justified. Perhaps the Authors could add some comments on this and make use of other rather simple statistical metrics to define the limit of 5% between natural and peri-urban catchments. An example could be to justify this value (of 5%) of imperviousness where H is estimated larger than 0.75 (for the R/S method), which is the mean between 0.5 (white noise behaviour at large scales) and ~1 (highly persistent).

We slightly modified the phrase in (page 7 of the annotated manuscript, line 11):

"Therefore, Figure 1 suggests that the analysed catchments might be classified as natural (U $\leq$ 5%), peri-urban (5%15%) catchments based on the corresponding range of values of *H*."

8) Ln. 196-197: "This suggests that the increase in impervious cover might cause more precipitation to bypass the groundwater storage", and Ln. 232-233: "Therefore, small catchments may not have sufficient water storage to influence the long-term dependence in flow time series."

The above sentences seem somehow contradictory, in the sense that if there is no clear relationship between H and catchment size because the catchment sizes are small compared to literature, then how come H is decreasing due to the increase of the imperviousness which has caused more precipitation to bypass the storage capacity? I believe the Authors try to explain this in Ln. 237-239 by mentioning that "Generally, catchments with lower rainfall totals and lower specific mean streamflow are found to have higher long-term dependence due to the longer dry weather periods and consequently longer low flow periods." However, only minor information is given for the rainfall totals and aridity of the examined catchments.

Maybe some further explanation here (and more information provided on the climatic conditions of the examined catchments) could help the Readers to better understand this point.

To clarify this point, we re-wrote the text at page 7 of the annotated manuscript (line 32), which now reads:

"These conflicting results may be due to the smaller size of the catchments used here compared to those in literature. For example, the size of the largest catchment used in this analysis (see Table 1) is comparable to the smallest of the catchments used by Szolgayova et al. (2014). The groundwater storage appears to be able to affect the persistence of the series of the less urbanized catchments irrespectively of the area. However, small urbanized catchments may not have sufficient water storage to influence the long-term dependence in flow time series, and an increase in imperviousness further limits the water storage capacity of the urban catchments. This may contribute to the lack of apparent relationship between the H exponent and the catchment size."

Spelling and grammar comments:

1) Ln. 187: "15% imperviousness". Please replace this with '15% of imperviousness'.
2) Ln. 192: "...due the high precipitation...". Please replace 'due the high' with 'due to high'.

Done.

I hope the Authors would find some of the above comments useful to their analysis.

We thank again Dr Dimitriadis for the thorough review and the suggestions, which have improved the manuscript.